# GLOBAL RELATIONAL MODELS OF SOURCE CODE

**Vincent J. Hellendoorn, Petros Maniatis, Rishabh Singh, Charles Sutton, David Bieber**
Google Research
{vhellendoorn,maniatis,rising,charlessutton,dbieber}@google.com

## ABSTRACT

Models of code can learn distributed representations of a program's syntax and semantics to predict many non-trivial properties of a program. Recent state-of-the-art models leverage highly structured representations of programs, such as trees, graphs and paths therein (*e.g.,* data-flow relations), which are precise and abundantly available for code. This provides a strong inductive bias towards semantically meaningful relations, yielding more generalizable representations than classical sequence-based models. Unfortunately, these models primarily rely on graph-based message passing to represent relations in code, which makes them de facto *local* due to the high cost of message-passing steps, quite in contrast to modern, global sequence-based models, such as the Transformer. In this work, we bridge this divide between global and structured models by introducing two new hybrid model families that are both global *and* incorporate structural bias: *Graph Sandwiches*, which wrap traditional (gated) graph message-passing layers in sequential message-passing layers; and *Graph Relational Embedding Attention Transformers* (GREAT for short), which bias traditional Transformers with relational information from graph edge types. By studying a popular, non-trivial program repair task, *variable-misuse identification*, we explore the relative merits of traditional and hybrid model families for code representation. Starting with a graph-based model that already improves upon the prior state-of-the-art for this task by 20%, we show that our proposed hybrid models improve an additional 10–15%, while training both faster and using fewer parameters.

## 1 INTRODUCTION

Well-trained models of source code can learn complex properties of a program, such as its implicit type structure (Hellendoorn et al., 2018), naming conventions (Allamanis et al., 2015), and potential bugs and repairs (Vasic et al., 2019). This requires learning to represent a program's latent, semantic properties based on its source. Initial representations of source code relied on sequential models from natural-language processing, such as $n$-gram language models (Hindle et al., 2012; Allamanis & Sutton, 2013; Hellendoorn & Devanbu, 2017) and Recurrent Neural Networks (RNNs) (White et al., 2015), but these models struggle to capture the complexity of source code.

Source code is rich in structured information, such as a program's abstract syntax tree, data and control flow. Allamanis et al. (2018b) proposed to model some of this structure directly, providing a powerful inductive bias towards semantically meaningful relations in the code. Their Gated Graph Neural Network (GGNN) model for embedding programs was shown to learn better, more generalizable representations faster than classical RNN-based sequence models.

However, the debate on effective modeling of code is far from settled. Graph neural networks typically rely on synchronous message passing, which makes them inherently *local*, requiring many iterations of message passing to aggregate information from distant parts of the code. However, state-of-the-art graph neural networks for code often use as few as eight message-passing iterations (Allamanis et al., 2018b; Fernandes et al., 2018), primarily for computational reasons: program graphs can be very large, and training time grows linearly with the number of message passes. This is in contrast to, *e.g.,* Transformer models (Vaswani et al., 2017), which allow program-wide information flow at every step, yet lack the powerful inductive bias from knowing the code's structure.

This leads us to a basic research question: *is there a fundamental dichotomy between global, unstructured and local, structured models?* Our answer is an emphatic no. Our starting point is the sequence-to-pointer model of Vasic et al. (2019), which is state-of-the-art for the task of localizing and repairing a particular type of bug. As a sequence model, their architecture can (at least potentially) propagate information globally, but it lacks access to the known semantic structure of code. To this end, we replace the sequence encoder of Vasic et al. (2019) with a GGNN, yielding a new *graph-to-mutlihead-pointer model*. Remarkably, this model alone yields a 20% improvement over the state of the art, though at the cost of being significantly larger than the sequence model.

Motivated by this result, we propose two new families of models that efficiently combine longer-distance information, such as the sequence model can represent, with the semantic structural information available to the GGNN. One family, the *Graph Sandwich*, alternates between message passing and sequential information flow through a chain of nodes within the graph; the other, the *Graph Relational Embedding Attention Transformer* (GREAT), generalizes the relative position embeddings in Transformers by Shaw et al. (2018) to convey structural relations instead. We show that our proposed model families outperform all prior results, as well as our new, already stronger baseline by an additional 10% each, while training both substantially faster and using fewer parameters.

## 2 RELATED WORK

**Distributed Representation of Programs:** There has been increasing interest in modeling source code using machine learning (Allamanis et al., 2018a). Hindle et al. (2012) model programs as sequences of tokens and use an $n$-gram model for predicting code completions. Raychev et al. (2015) use conditional random fields (CRFs) to predict program properties over a set of pairwise program features obtained from the program's dependency graph. Many approaches use neural language models to embed programs as sequences of tokens (Bhoopchand et al., 2016; White et al., 2015). Some techniques leverage the ASTs of programs in tree-structured recurrent models (Piech et al., 2015; Parisotto et al., 2016; Chen et al., 2018). code2vec (Alon et al., 2018) and code2seq (Alon et al., 2019) model programs as a weighted combination of a set of leaf-to-leaf paths in the abstract syntax tree. Finally, Allamanis et al. (2018b) proposed using GGNNs for embedding program graphs consisting of ASTs together with control-flow and data-flow edges. Some recent models of code embed run-time information of programs, *e.g.,* program traces, besides syntactic information (Wang et al., 2018). In this paper, we explore the space of combining sequence-based and graph-based representations of programs, as well as introduce a Transformer-based model with additional program-edge information to learn program representations. Fernandes et al. (2018) also combine an RNN and a GNN architecture, achieving slight improvements over a GGNN. However, they only consider a single RNN layer inserted at the start; we include larger and more diverse hybrids, as well as entirely different combinations of structural and sequential features.

**Neural Program Repair:** Automatically generating fixes to repair program bugs is an active field of research with many proposed approaches based on genetic programming, program analysis and formal methods, and machine learning (Monperrus, 2018; Gazzola et al., 2019). In this paper, we focus on a specific class of repair task called VarMisuse as proposed by Allamanis et al. (2018b), who use a graph-based embedding of programs to predict the most likely variable at each variable-use location and generate a repair prediction whenever the predicted variable is different from the one present, using an enumerative approach. Vasic et al. (2019) improved this approach by jointly predicting both the bug and repair locations using a two-headed pointer mechanism. Our multi-headed pointer graph, graph-sandwich and GREAT models significantly outperform these approaches.

## 3 SEMI-STRUCTURED MODELS OF SOURCE CODE

Models of code have so far either been structured (GNNs) or unstructured (RNNs, Transformers). Considering the graph-based models' substantially superior performance compared to RNNs despite their locality limitations, we may ask: to what extent could global information help GNNs, and to what extent could structural features help sequence-based models?

### 3.1 MODELS

We address the questions of combining local and global information with two families of models.

**Graph-sandwich Models** Let $T = \langle t_1, t_2, \cdots, t_n \rangle$ denote a program's token sequence and $\mathcal{G} = (\mathcal{V}, \mathcal{E})$ denote the corresponding program graph, where $\mathcal{V}$ is the set of node vertices and $\mathcal{E}$ is the list of edge sets for different edge types. In both graph and sequence based models, the nodes $v$ maintain a state vector $h^{(v)}$ that is initialized with initial node embedding $x^{(v)} \in \mathbb{R}^D$.

In a GGNN layer, messages of type $k$ are sent from each node $v \in \mathcal{V}$ to its neighbors computed as $m_k^{(v)} = \text{LinearLayer}_k(h^{(v)})$. After the message passing step, the set of messages at each node are aggregated as $m^{(v)} = \Sigma_{e_k(u,v) \in \mathcal{E}} \; m_k^{(u)}$. Finally, the state vector of a node $v$ is updated as $h_{\text{new}}^{(v)} = \text{GRU}(m^{(v)}, h^{(v)})$. In an RNN layer, the state of a node $v$ (corresponding to a terminal leaf token $t_i$ in $T$) is updated as $h_{\text{new}}^{(v)} = f(t_v, h^{(t_{i-1})})$. A Transformer compute $t_v \rightarrow \mathbf{q_t}, \mathbf{k_t}, \mathbf{v_t}$, corresponding to a query, key and value for each token.[1] Each token then computes its attention to all other tokens using $e_{ij} = (\mathbf{q_i}\mathbf{k_j}^\top)/\sqrt{N}$,[2] which can be soft-maxed to yield attention probabilities $a_{ij} = \exp(e_{ij})/\Sigma \exp(e_{i,:})$.

Our first class of models follows from the observation that $T \subseteq V$; *i.e.,* the source code tokens used by sequence models, like RNNs, are by definition also nodes in the program graph, so GGNNs update their state with every message pass. We can thus envision a combined model that uses each of these as a building block; for instance, assuming initial node features $x^{(v)} \in \mathbb{R}^D$, the formula [RNN, GGNN(3), RNN] describes a model in which we first run an RNN on all tokens $\in T$ (in lexical ordering), then, using these as initial states for $v \in T$ while using the default node-type embeddings for all other nodes, run three message passing steps using a GGNN, after which we again gather the nodes corresponding to $T$ and update their state with an additional RNN pass.

The resulting family of models alternates GGNN-style message passing operations and layers of sequence-based models. By varying the number and size of sequential layers and blocks of GGNN-style message passing, this variant particularly provides insight into the first question above (how can global information help GNNs?), by showing the transition in performance potential of models that increasingly incorporate sequential features. We refer to this class of models as *sandwich models*.

**Graph Relational Embedding Attention Transformer** The above family of models still rely on explicit message passing for their structural bias, thereby only indirectly combining structural and global information to the model. We may wish to instead directly encode structural bias into a sequence-based model, which requires a relaxation of the 'hard' inductive bias from the GGNN. For Transformer-based architectures, Shaw et al. (2018) show that relational features can be incorporated directly into the attention function by changing the attention computation to:[3] $e_{ij} = (\mathbf{q_i} + b_{ij})\mathbf{k_j}^\top/\sqrt{N}$ where $\mathbf{q_i}$ and $\mathbf{k_j}$ correspond to the query and key vectors as described above, $b_{ij}$ is an added bias term for the specific attention weight between tokens $i$ and $j$, and $N$ is the per-head attention dimension. In our case, we compute $b_{ij} = W_e^\top \mathbf{e} + b_e$, where $W_e \in \mathbb{R}^N, b_e \in \mathbb{R}$, and $\mathbf{e} \in \mathbb{R}^N$ is an embedding of the edge type connecting nodes $i$ and $j$, if any. If multiple edge types are present between two nodes, the resulting biases are simply added. We name this model GREAT, for Graph Relational Embedding Attention Transformer.

### 3.2 ARCHITECTURAL DETAILS

In this section, we present details of different architectures we compare and their hyperparameters.

**General:** All of our models follow the structure proposed by Vasic et al. (2019), stacking an initial token-embedding layer, a 'core' model that computes a distributed representation of the code under inspection (in their case, an LSTM), followed by a projection into two pointers for the localization and repair tasks (see Section 4). This core model is the part varied in our work. We use

---

[1]Transformers introduce several more components, including multi-headed attention, feed-forward blocks in every layer and layer-normalization (Vaswani et al., 2017).

[2]Where the $\sqrt{N}$ term is used to scale the attention weights

[3]Note that, although equivalent, Shaw et al. (2018) add the relational bias to the 'key' computation instead.

`SubwordTextEncoder` from Tensor2Tensor (Vaswani et al., 2018) to generate a 10K sub-token vocabulary from training data and embed each token by averaging embeddings of its sub-token(s).

**GGNN:** Many types of graph-based message-passing neural networks have been proposed for code, mostly differing in how a node's state is updated based on 'messages' sent by nodes it is connected to. Most commonly used is the gated graph neural network (GGNN) (Li et al., 2015), which uses a GRU cell (Cho et al., 2014) to update a node's state. Although other options sometimes outperform this architecture, the improvements are generally minor, so we rely on this model for our baseline. One hyperparameter of the architecture is whether to use different transformations at each message-passing step, or to reuse one set of transformations for multiple message passes. Following Allamanis et al. (2018b), we use blocks of two message-passing layers, in which the first layer is repeated three times, for four message passes per block. We then sweep over GGNN architectures that repeat these blocks 1 to 4 times (thus yielding 4 to 16 message passes). By default, the message dimension is set to 128, but we include an ablation with 256-dimensional messages as well.

**RNNs:** We experimented with the one-directional entailment-attention-based RNN proposed by Vasic et al. (2019), but found a simpler bi-directional RNN architecture to work even better. We use GRUs as the recurrent cells, vary the number of layers from 1 to 3, and the hidden dimension (of the concatenated forward and backward component) in {128, 256, 512}.

**Transformers:** We base our architecture on the original Transformer (Vaswani et al., 2017), varying the number of layers from 1 to 10 and the attention dimension in {128, 256, 512, 1024}.

**Sandwich Models:** We distinguish between two types of sandwich models: 'small' sandwiches, which add a single RNN or Transformer to a GGNN architecture, and 'large' sandwiches, which wrap every message-passing block (as defined above) with a 128-dimensional (bi-directional) RNN/Transformer layer. We vary the number of message-passing blocks from 1 to 3 (corresponding to 4 to 12 message passes) to span a similar parameter domain as the GGNNs above (ca. 1.5M – 5M), increasing the number of layers to 2 and their dimension to 512 for a later ablation.

**GREAT:** Uses the same architectural variations as the Transformer family; edge-type embedding dimensions are fixed at the per-head attention dimension, as described above.

**Global hyper-parameters:** We train most of our models with batch sizes of {12.5K, 25K, 50K} tokens, with the exception of the Transformer architectures; due to the quadratic nature of the attention computation, 25K tokens was too large for these models, so we additionally trained these with 6.25K-token batches.[4] Learning rates were varied in {1e-3, 4-e4, 1e-4, 4e-5, 1e-5} using an Adam optimizer, where we omitted the first option for our GGNN models and the last for our RNNs due to poor performance. Sub-tokens were embedded using 128-dimensional embeddings.

**Hardware:** all our models were trained on a single Tesla P100 GPU on 25 million samples, which required between 40 and 250 hours for our various models. However, we emphasize that overall training time is not our main objective; we primarily assess the ultimated converged accuracy of our models and present training behavior over time for reference of our various models' training behavior.

## 3.3 ABOUT GRAPH REPRESENTATIONS OF CODE

Our program graphs borrow many edge types from Allamanis et al. (2018b), such as data-flow (*e.g.,* read & write), adjacent-token, and syntactic edges, which we further augment with edges between control-flow statements and function calls. When representing programs as graphs, a key decision needs to be made regarding the Abstract Syntax Tree (AST). Typically, one of the edge types in the graphs represents syntactic parent-child relationships in the AST. Additionally, some of the edges representing relations (*e.g.,* control-flow) are naturally represented as edges between internal nodes in this tree, *e.g.,* between two `IfStatement` nodes. However, ablations often find that the effectiveness of including the AST is limited in graph-based models (Allamanis et al., 2018b).

This raises the question of whether it is possible to represent programs as graphs that include sequential and semantic information, but not syntax. To this end, we propose a *leaves-only* graph representation for code as follows: edges that represent semantic relationships such as control flow and data flow can easily be moved down from internal nodes – which typically represent a span of

---

[4]Which still translates into 50+ samples per batch on average

multiple tokens – to those leaf nodes in the graph that represent the begin token of that span. Thus, an edge that used to connect two `IfStatement` interior AST nodes is moved down to connect the corresponding `if` tokens. Now, the AST can be omitted entirely, thereby removing parent-child relations among syntax nodes, producing what we call a leaves-only graph. This latter representation is substantially more compressed than the graphs with ASTs, often using 2–3x fewer nodes (while retaining most of the edges), and additionally aligns better with sequence-based models, because all edges are directly connected to the original code tokens.[5] We compare both settings for each graph-based model, but unless otherwise specified, we use the 'full' graphs for the regular GGNN model and the 'leaves-only' graphs (without ASTs) for the sandwich and GREAT models.

## 4 EXPERIMENTAL SETUP

**The VarMisuse Task**  We focus our study on the variable-misuse localization-and-repair task (Vasic et al., 2019): given a function, predict two pointers into the function's tokens, one pointer for the location of a variable use containing the wrong variable (or a special no-bug location), and one pointer for *any* occurrence of the correct variable that should be used at the faulty location instead.

**Synthetic Dataset**  We used the ETH Py150 dataset (Raychev et al., 2016), which is based on GitHub Python code, and already partitioned into train and test splits (100K and 50K files, respectively). We further split the 100K train files into 90K train and 10K validation examples and applied a deduplication step on that dataset (Allamanis, 2018). We extracted all top-level function definitions from these files; any function that uses multiple variables can be turned into a training example by randomly replacing one variable usage with another. As there may be many candidates for such bugs in a function, we limit our extraction to up to three samples per function to avoid biasing our dataset too strongly towards longer functions. For every synthetically generated buggy example, an unperturbed, bug-free example of the function is included as well, to keep our dataset balanced, yielding ca. 2M total training and 755K test samples. Finally, we train with functions with up to 250 tokens; at test time, we raise this to 1,000 to study our models' generalization to longer functions.

**Metrics**  As we are mainly interested in contrasting the behavior of different models of code, we focus most of our results on the various models' learning curves by tracking development-set accuracy on 25K held-out samples every 250K samples as the models train. Here, we measure two accuracy metrics: localization accuracy (whether the model correctly identifies the bug's location for buggy samples); and (independently) repair accuracy (whether the model points to the correct variable to repair the bug). Note that these metrics focus on buggy samples; the models also determine whether a function is buggy, which we discuss below. We group all models by their 'family', as categorized in Section 3.2, reporting the maximum held-out performance per family.

For deeper insight into the fully trained models' performance, we also analyze the performance of the best models in each family in more depth on the test portion of our synthetic dataset. Specifically, we assess their bugginess-classification accuracy (whether the model correctly identifies the method as (non-)buggy) and their joint localization and repair accuracy (for buggy samples, how often the model correctly localizes and repairs the bug). Here, we also increase the maximum function size to 1,000 tokens and analyze the impact of longer functions on our models' performance.

### 4.1 DATA & CODE RELEASE

We release a public implementation of the GREAT model based on Tensorflow, as well as the program graphs for all samples in our training and evaluation datasets whose license permits us to redistribute these at: `https://doi.org/10.5281/zenodo.3668323`, which tracks the latest release of our Github repository at: `https://github.com/VHellendoorn/ICLR20-Great`.

## 5 RESULTS

There are many degrees of freedom in our family of models, so we structure our results around a series of comparisons, which we analyze and discuss in this section. We start with our key re-

---

[5]Which, we conjecture, improves the interaction between the two types of models.

sult, which compares all our model families (RNNs, Transformers, GGNNs, Sandwich hybrids, and GREAT models) across a comparable parameter domain (ca. 1.4M – 5.5M parameters) in Figure 1.

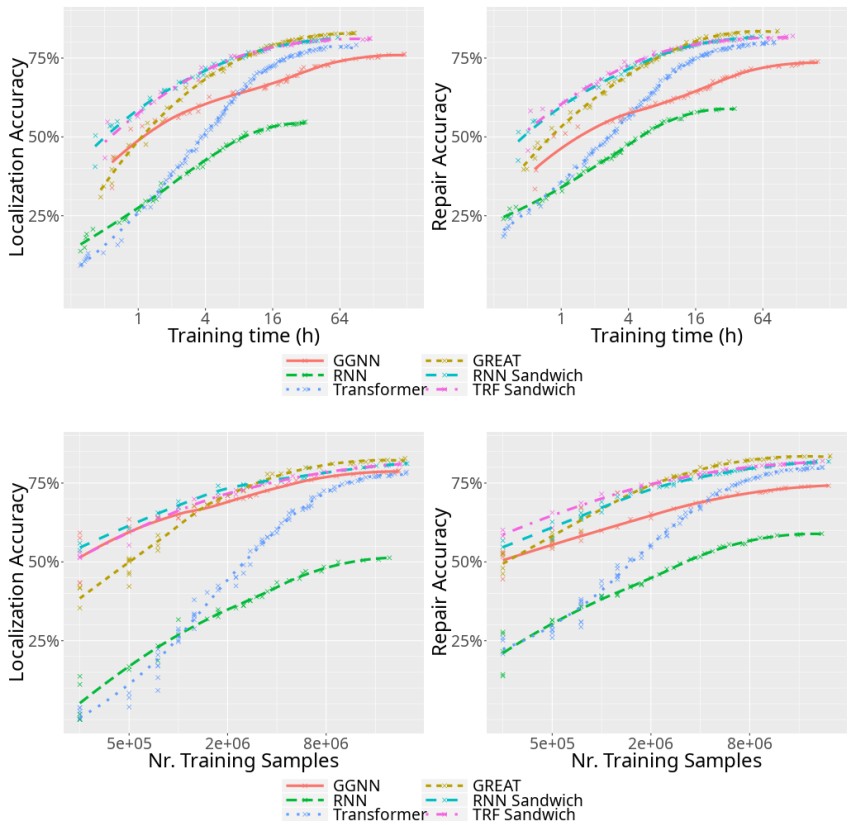

Figure 1: Comparison of top-performing models from all model families across a comparable parameter domain of 1.5M – 5M parameters. Performance visualized using the localization (left) and repair (right) accuracy Pareto fronts w.r.t. both training time (top) and number of training samples seen (bottom), both log-scaled, where an epoch is ca. 2M samples. In all cases, our proposed models substantially outperform GGNNs from early in the training process.

Although there are subtle differences between the models' behavior on localization and repair accuracy, the overall picture is consistent: whereas our newly proposed graph-to-multihead-pointer models already substantially outperform RNN-based models (the previous state-of-the-art), and sometimes Transformers, the hybrid global & structured models achieve significantly better results faster.

Time-wise (the top two figures), the GGNN models take the longest to converge, continuing to improve slightly even after a week of training mainly because their largest (16-layer) architecture starts to dominate the 12-layer version's performance after ca. 170h. The Sandwich models follow its training curve, but are more accurate and faster to converge, achieving especially good results for limited training budgets, partly because they succeeded with just 4 – 8 layers of message passing by relying on their global components.

The Transformer architecture, although slower at first, widely outperforms the RNN as a baseline model. The GREAT model tracks its learning curve, starting out slower than the models with explicit message passing, but gradually overtaking them after ca. 10h, as the underlying Transformer becomes increasingly effective. We note that this model achieves state-of-the-art results despite having, at the time of this writing, received less training time (ca. 64h compared to up to 240h).

The bottom half of Figure 1 abstracts away the potentially confounding issue of implementation speed of our models by tracking performance w.r.t the number of training samples. Naturally, the end-points of the curves (the converged accuracy) are identical, but importantly the various models'

training curves are quite similar; even here, the GGNN is only able to outperform some of our proposed models briefly, yielding inferior performance to all within just one epoch.

The RNN and GGNN models appear to be particularly complementary; even though the RNN's localization accuracy is very poor compared to the GGNN, the combined model still sustains a ~5% improvement on the latter.[6] However, the Transformer Sandwich does not seem to benefit similarly, showing virtually no difference in performance with the RNN Sandwich model. This strongly suggests that the Transformer's ability to access long-distance information overlaps in large part (though not entirely, given GREAT's performance) with the GGNNs' ability to do so using message passing. We conjecture that the Transformer learns to infer many of the same connections (*e.g.,* data-flow, control-flow) that are encoded explicitly in the graph's message passing.

To understand the behavior of the many models and combinations that may be used for code, we now explore the variations on our choices of parameters, models, and metrics.

## 5.1 LARGER MODELS

Model capacity is a potential threat to any comparison between models of different families. We aimed to ensure a fair comparison in the previous section by selecting a range of hyper-parameters (which includes the number of stacked layers) for these architectures that span a similar parameter count range. For instance, a 6-layer Transformer with 512-dimensional attention is comparable to a 2-layer 512-dimensional bi-directional RNN and an 8-layer GGNN. However, all these architectures are relatively modest, having at most ~5M parameters. By increasing the number, and dimensionality of their layers, we can evaluate a second family of models with ca. 5–20M parameters.

Figure 2 shows the performance for the low- and high-parameter variations for each of our best-performing model families. Overall, while providing more parameters to the GGNNs made virtually no difference, all our hybrid models increase 2–3% in both localization and repair accuracy, providing further support for combining global, structured models. The best-performing instances of each model family were consistently the larger architectures, 15M parameters for the GGNN, 12.5M & 10M for the RNN and Transformer Sandwiches respectively and 7.9M for GREAT.[7]

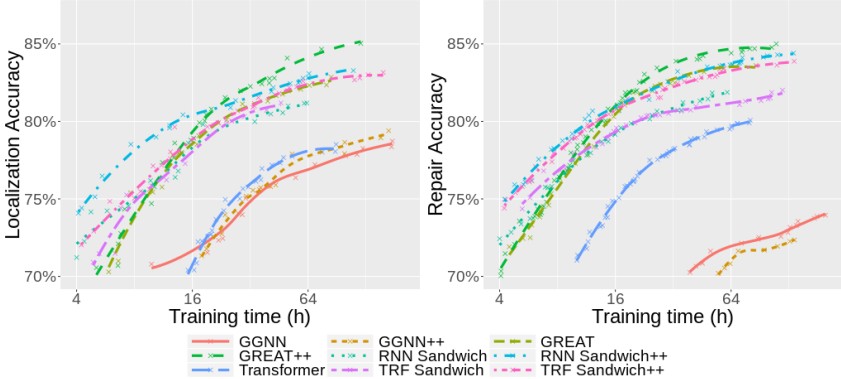

Figure 2: Comparison of smaller (1.5–5M parameter) and larger (5–20M parameter, identified with '++') variants of each model family. Localization and repair accuracy Pareto-front w.r.t. time.

## 5.2 ON SYNTACTIC INFORMATION

In the previous results, the GGNN models were trained on 'full' graphs (as described in Section 3.3), that use the code's AST structure, and the sandwich models on 'leaves-only' graphs, with only source token nodes, and edges moved to connect these directly. These settings are arguably each appropriate to the underlying model, but both models can also use the alternative setting.

---

[6]Difference up to convergence of the combined model; the accuracy gap shrinks to 2.1% after ca. 10 days.
[7]Which achieves state-of-the-art results in all settings despite having had comparatively less training time.

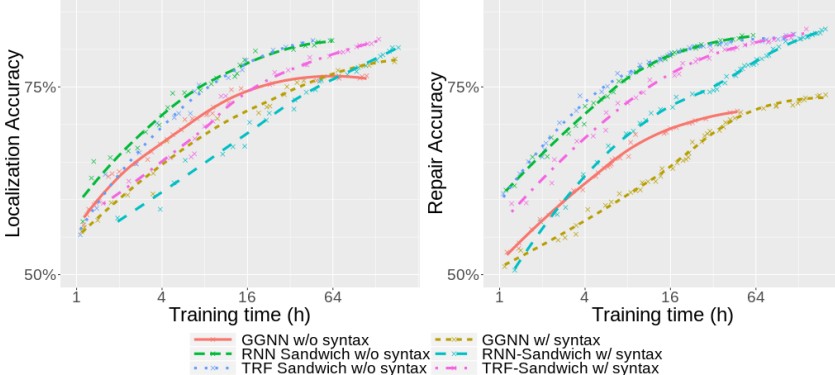

Figure 3: Comparing impact of graph representation on GGNNs and RNN Sandwich models. Localization and repair accuracy Pareto-front w.r.t time. Note: y-axis cropped to simplify comparison.

Figure 3 shows the training curves for the alternative settings. In all cases, the models that do not use syntax train substantially faster because each sample's graph representation is more than twice as small, so these models naturally lead in accuracy early on in training. However, whereas the GGNN equipped with syntax overtakes its counter-part within ca. 48h, the sandwich models display a much longer lag, with no cross-over observed at all on localization accuracy in this time window.[8]

The sandwich model on full graphs still compares favorably with the GGNN baseline, though its early training behavior is not as effective. It is also interesting to note that the best-performing Sandwich models in this setting were consistently architectures with more message-passing steps. This may be due to the additional distance between information propagated along the tokens and along semantic edges, which in this setting are almost universally connected to AST-internal nodes.

## 5.3 RNNs IN SANDWICHES: SINGLE VS. MANY

Recent work on neural summarization also mixed RNNs and GGNNs (Fernandes et al., 2018), but did so by inserting a single RNN layer into a GNN architecture, before any message passing. We compare this architecture to a full Sandwich model in Figure 4. Although a single RNN certainly helps compared to the GGNN, interleaving RNNs and GGNN-style message passes performed substantially better. In fact, the best performing full Sandwiches used *fewer* parameters than the Single models because they used 8 message passes instead of 12, relying more heavily on the RNNs.[9]

## 5.4 TEST-SET ANALYSIS

Having identified our best-performing models in each family, we now study their performance on the test data, specifically using the metrics used in Vasic et al. (2019) (see Section 4) in Table 1. In general, the two metrics correlated well; models that accurately determined whether a function contained a bug also accurately identified the bug (and repair), as may be expected. The baseline RNN model achieves a modest 44% accuracy at the latter task; this is slightly lower than reported in prior work (Vasic et al., 2019), which is likely due in part to our dataset de-duplication. Transformers and GGNNs perform substantially better (and comparably, though the latter trained 6x longer for this performance), but still fall well short of our hybrid models' performance, which are especially much more accurate on long functions. The GREAT model shows most promise on the repair task, already outperforming the sandwich models despite having so far had limited training time.

---

[8]Given the slope of the two curves, we may expect a reversal after 10+ days of training.

[9]The same pattern held for high-parameter versions of these models, where the performance gap also grew.

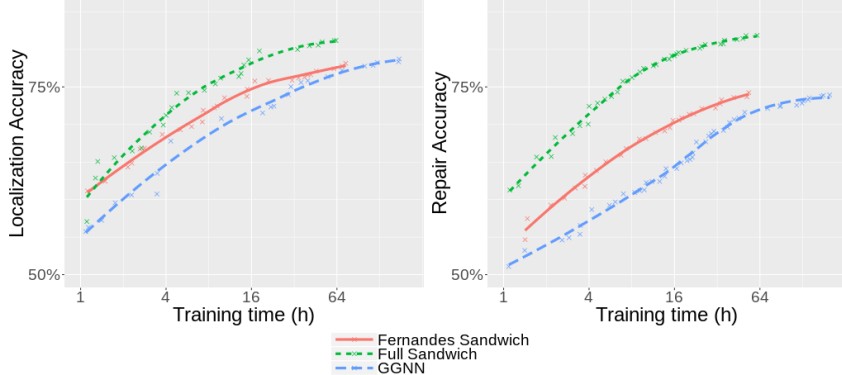

Figure 4: Comparison of RNN sandwiches with a single RNN vs. those with RNNs around every message-passing block ('Multi'). Localization and repair accuracy Pareto-front w.r.t. time.

Table 1: Test-set results of best-performing models by family, on metrics of Vasic et al. (2019). Grouped by maximum length; 6.5% of test set samples exceeded 250 tokens (the training limit).

| Model Family | Class. Accuracy | | Loc & Rep Accuracy | | Parameters | Training Time |
|---|---|---|---|---|---|---|
| | $\leq 250$ | $\leq 1000$ | $\leq 250$ | $\leq 1000$ | | |
| RNN[1] | 71.8% | 70.6% | 44.4% | 42.5% | 4.3M | 31.3h |
| Transformer | 75.9% | 73.2% | 67.7% | 63.0% | 3.7M | 41.5h |
| GGNN | 81.4% | 79.2% | 64.0% | 60.9% | 5.5M | 241h |
| RNN Sandwich | **82.5%** | **81.9%** | 75.8% | **73.8%** | 12.6M | 109h |
| Transformer Sandwich | 81.1% | 78.1% | 74.5% | 71.4% | 10M | 161h |
| GREAT | 80.1% | 76.9% | **76.4%** | 73.1% | 7.9M | 120h |

1: a stronger version of the model proposed in Vasic et al. (2019) (previous SOTA).

## 5.5 REAL BUGS ANALYSIS

We want to ensure that our models can be useful for real bugs and do not simply overfit to the synthetic data generation that we used. This risk exists because we did not filter our introduced bugs based on whether they would be difficult to detect, for instance because they conflate variables with similar names, usage, or data types; presumably, such bugs are more likely to escape a developer's notice and find their way into real code bases. It is therefore expected that performance on real-world bugs will be lower for all our models, but we must assert that our proposed models do not just outperform GGNNs on synthetic data, e.g. by memorizing characteristics of synthetic bugs.

To mitigate this threat, we collect real variable misuse bugs from code on Github. Specifically, we collect ca. 1 million commits that modify Python files from Github. We extracted all changes to functions from these commits, filtering these according to the same criteria that we used to introduce variable-misuse bugs: we looked for commits that exclusively changed a single variable usage in a function body from one variable in scope to another. We focus on functions with up to 250 tokens, since all our models performed substantially better on these in Table 1. We identified 170 such changes, in which we assumed that the version before the change was buggy and the updated version correct. We removed any functions that had occurred in our training data, of which we found 9 and paired the remaining functions up (correct and buggy) to create a real-world evaluation set with 322 functions, which we presented to our models.

Table 2 shows the results of running our various models on these bugs. In general, these were clearly substantially more difficult for all our models than the synthetic samples we generated. However, we see a clear difference in performance with all our proposed models performing substantially better than previous baselines, and showing favorable precision/recall trade-offs.

Table 2: Results on 322 paired buggy and non-buggy samples from 161 real variable misuse bugs mined from Github commits. 'Class.' measures accuracy at identifying non-buggy samples; 'Prec.' and 'Rec.' capture the precision and recall at identifying the correct localization and repair on the buggy samples.

| | *All Samples* | | | Precision at | Precision at |
|---|---|---|---|---|---|
| **Model Family** | Class. | Prec. | Rec. | Recall = 5% | Recall = 10% |
| RNN | 52.8% | 13.3% | 46.6% | 44.4% | 23.5% |
| Transformer | 62.1% | 15.8% | 47.2% | 33.3% | 17.6% |
| GGNN | 65.8% | 17.7% | 42.2% | 44.4% | 23.5% |
| RNN Sandwich | 69.6% | **28.6%** | **43.5%** | **77.8%** | **64.7%** |
| Trans. Sandwich | **75.2%** | 21.5% | 40.4% | 33.3% | 35.3% |
| GREAT | 70.2% | 23.7% | 36.7% | 44.4% | 29.4% |

## 6 CONCLUSION

We demonstrate that models leveraging richly structured representations of source code do not have to be confined to local contexts. Instead, models that leverage only limited message passing in combination with global models learn much more powerful representations faster. We proposed two different architectures for combining local and global information: sandwich models that combine two different message-passing schedules and achieve highly competitive models quickly, and the GREAT model which adds information from a sparse graph to a Transformer to achieve state-of-the-art results. In the process, we raise the state-of-the-art performance on the VarMisuse bug localization and repair task by over 30%.

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
