# OpenReview forum: "Global Relational Models of Source Code"
_ICLR.cc/2020/Conference — Accept (Poster)_

### Official Review · AnonReviewer3 · 2019-10-21
**Official Blind Review #3**

**Rating:** 6

**Review:**

In this paper, the authors proposed a new method to model the source code for the bug repairing task. Traditional methods use either a global sequence based model or a local graph based model. The authors proposed a new sandwich model like [RNN GNN RNN]. The experiments show that such simple combination of models significantly improve the localization and repair accuracy.

The idea is simple, so the technical contribution is a bit low. But the message is clear. The sandwich model can benefit from GNN model that can achieve a higher accuracy at the beginning of training where transformer did a poor job. At the end of training, the sandwich model outperforms both kinds of models.

Here are some detailed comments:
1. It would be interesting to have the complete result from Transformer in Figure 1 and Figure 2 which is missing.
2. The results from GGNN (smaller model) in Figure 1 and Figure 2 seems to be not the same.
3. One major benefit of GNN is its efficient local computation. Some industrial applications have also used GNN for recommendation that can be trained very fast. Why GGNN is so slow in this paper? Is this because of implementation?


**Experience Assessment:**

I have published one or two papers in this area.

**Review Assessment: Checking Correctness Of Derivations And Theory:**

I assessed the sensibility of the derivations and theory.

**Review Assessment: Checking Correctness Of Experiments:**

I assessed the sensibility of the experiments.

**Review Assessment: Thoroughness In Paper Reading:**

I read the paper at least twice and used my best judgement in assessing the paper.

---

> ### Author Response · Authors · 2019-11-09
> **Response to AnonReviewer3**
>
> We thank the reviewer for their constructive comments. With regards to the detailed comments:
>
> 1. We have updated the paper with the now-completed experimental results, which are largely in line with was submitted, but mainly demonstrate that our GREAT model outcompetes the Sandwich models in some metrics in the long term.
>
> 2. The difference in curvature is due to the LOESS smoothing picking up smaller variations on this substantially zoomed in region of the data (the Y-axis is zoomed in the region of 70-85% in Fig. 2 compared to unzoomed y-axis in Fig. 1); the underlying data points (marked with ‘x’) are the same. The cause for this transition is the underlying GGNN; at this inflection point, the 12-layer model first starts to out-compete the 8-layer architectures, setting a series of new highs in quick succession. A similar, if more nuanced inflection point can be seen at around 100h, where the 16-layer GGNN overtakes it in turn.
>
> 3. There are several perspectives on this. From a performance point of view, the GGNN in source code settings is somewhat hindered by the many distinct types of edges that are present there (more than 20 in our work), which roughly linearly increase modeling time. We note that this is not out of line with prior work on modeling source code, which has invested considerable efforts in making these models scale to moderate-sized training sets (see Allamanis et al., ICLR’18 & Balog et al., ICLR’19). More generally, however, we stress that the speed of the models is not our main target here; even if we compare the performance of these models w.r.t. the number of samples seen, regardless of how long it took to process these, we still see very similar training curves with the same ultimate outcomes. We recognize that our plots’ emphasis of time (on the x-axis) may have created the wrong impression and have therefore added another, time-agnostic comparisons to Figure 1 in the paper, in which we instead compare the number of samples seen to training accuracy. Here too, both in the limit and throughout the training process, our models substantially outperform the prior state-of-the-art.

---

### Official Review · AnonReviewer1 · 2019-10-22
**Official Blind Review #1**

**Rating:** 3

**Review:**

Strength:
-- Interesting problem
--The paper is well written and easy to follow
-- The proposed approach seems very effective

Weakness:
-- the novelty of the proposed is marginal
-- Some of the claims are not right in the paper

This paper studied learning the representations of source codes by combining sequential-based approaches (RNN, Transformers) and graph neural network to model both the local and global dependency between the tokens. Experimental results on both synthetic and real-world datasets prove the effectiveness of the proposed approach.

Overall, the paper is well written and easy to follow. However, the novelty of the proposed technique seems to be marginal to me. Some of the claims in the paper are not right. In the abstract, the authors said the graph neural network is more local-based while the transformer is more global-based. The essential difference between the two approaches lie in the way of constructing the graphs since transformer used the fully-connected graph (more local dependency) while graph neural networks usually capture the long-range dependency.

And there are actually some existing work that have already explored this idea in the context of natural language understanding, e.g.,
Contextualized Non-local Neural Networks for Sequence Learning. https://arxiv.org/abs/1811.08600
The authors should clarify the difference between these approaches.



**Experience Assessment:**

I have published in this field for several years.

**Review Assessment: Checking Correctness Of Derivations And Theory:**

I carefully checked the derivations and theory.

**Review Assessment: Checking Correctness Of Experiments:**

I assessed the sensibility of the experiments.

**Review Assessment: Thoroughness In Paper Reading:**

I read the paper at least twice and used my best judgement in assessing the paper.

---

> ### Author Response · Authors · 2019-11-09
> **Response to AnonReviewer1**
>
> We appreciate your comments and are glad that you found the paper easy to follow and the results to be noteworthy.
>
> First of all, we would like to emphasize that the main novel contribution of our paper is to present two new code-embedding models (Sandwich and GREAT), which achieve significantly better performance and accuracies than GGNNs (Allamanis et al. 2018), which is considered to be the current state-of-the-art for modeling code.
>
> Secondly, this performance follows from our key conceptual contribution, which is the insight that, while GNNs succeed by capturing information that is not _lexically_ local, the choice of edge types and message passing still confines it to a _topologically_ local space. That is, although a GNN can reach information from far away, it is heavily biased towards information that is reachable via relatively few ‘hops’ in the graph for entirely practical reasons (i.e., because it’s very expensive to run more than, say, 12 message-passing steps). This is in sharp contrast to e.g. Transformer models, which are ‘global’ in that they can choose to attend to any information at any point (even across tokens that may be far more than, say, 12 hops away along any program-graph edge paths). We demonstrate through our various proposed hybrid models that although this structural bias does give the GNN an initial edge over unstructured counter-parts, its lack of truly global information is also a limitation that can be complemented, and the model greatly improved, by combination with other, global models. This ties in closely to the claim (in the abstract) of graph-based models being ‘local’; this is by no means a mistake, but rather an essential insight to understand our contribution and proposed models. We will clarify this in the paper.
>
> We appreciate your reference to the related work on “Contextualized Non-local Neural Networks for Sequence Learning”; we read the paper with interest. The two methods are similar in spirit, but are architecturally rather different. Our GREAT model is a direct extension to the Transformer architecture, whereas the previous work makes a number of architectural changes (such as a different similarity measure from Transformers, explicit layers for learning graph structure and performing message passing, concatenating edge features, updating node states via gating, etc.), whose effect could be challenging to disentangle. Another key difference is that in our domain, programs have much more explicit and deterministic structure, such as control flow, data flow, compared to natural language. This leads to a much larger set of edge types than used in that work.
>
> Additionally, our sandwich models present a very different point in the architectural-design space, where the graph structure is fixed a-priori from the program structure and allows for additional layer alterations between the Graph and RNN layers, whereas this work only applies a single LSTM (or CNN) before any message passing, much like Fernandes et al. (ICLR’19). We will add this discussion to the related work section.
>
> Please let us know if this clarifies your concerns.

---

> > ### Author Response · Authors · 2019-11-13
> > **Follow-up to AnonReviewer1**
> >
> > Please let us know if our response helped clarify your concerns regarding the novelty of our sandwich and GREAT models, and correctness of the claims. Also, please let us know about any additional clarifications or questions that may help improve your assessment of our work.

---

### Official Review · AnonReviewer2 · 2019-10-23
**Official Blind Review #2**

**Rating:** 6

**Review:**

The paper proposes improvements on existing probabilistic models for code that predicts and repairs variable misuses. This is a variant of the task, proposed by Vasic et al. The task takes a dataset of python functions, introduces errors in these functions and makes a classifier that would identify what errors were introduced and effectively reconstruct the original code.

The paper claims to improve state-of-the-art results published by Vasic et al for this task, however the RNN model by Vasic et al was known to be far from optimal when the work was published. Furthermore, that task was evaluated on artificially introduced changes (the original code could contain an error), but it is not clear that the improvements would have any practical effect. In fact, I conjecture that the bug-detector is in fact worse, because the entire dataset is not sufficiently large for millions of parameters and it is not clear that bugs that were originally the dataset were not learned by the better model, making it worse at spotting them. Given the relatively thinner contribution on the rest of the paper, I think this would be a valid question to be addressed to show the effectiveness of the model beyond accuracy on the artificial task.

The paper does a number of contributions to the neural architecture. The most important change precision-wise is to use transformer model instead of RNN (the model used by Vasic et al).  This change is also what makes the work perform as well or better than GGNN-based approaches. The paper then proposes to improve the transformer model by modifying the attention where there are edges. The rest of the contributions seem to be addressing the problem of aster convergence speed.  The other contribution of the paper is by selecting which edges to include and it is also shown to improve convergence speed.

Given that most of the work talks about performance, it would also help if the authors clarify what kind of hardware was used and which optimizer.

Q: Why a larger transformer model was not evaluated?

More minor issues:
“We conjecture that the Transformer learns to infer many of the same connections”. There is no confirmation for this besides similar accuracy, but if this is the case, why would I change the architecture and not just try with initializing the vectors to values corresponding to this knowledge and get faster convergence?
page 3, “where q and k correspond to the query and key vectors as described above,”. It seems it is q_i and k_j?

Update after the rebuttal:

 - I thank the authors for running additional experiments on a short notice. I have some reservations about their correctness though (we still do not know if there were bugs fixed in these commits, their number is very low and the authors seem to have cherrypicked the numbers to show - their first updated revision had recall at 20%, then decided to show it at 10%, the RNN baseline is actually having the highest recall of all although it is not highlighted). I am not sure that data cleaning of this small evaluation sample will not show a different picture.

 - I actually increase my score a bit (I was torn in the beginning), because this is one of the first papers to run transformer model on code. I still think the actual contributions of the paper are minor.

**Experience Assessment:**

I have published one or two papers in this area.

**Review Assessment: Checking Correctness Of Derivations And Theory:**

I carefully checked the derivations and theory.

**Review Assessment: Checking Correctness Of Experiments:**

I carefully checked the experiments.

**Review Assessment: Thoroughness In Paper Reading:**

I read the paper thoroughly.

---

> ### Author Response · Authors · 2019-11-09
> **Response to AnonReviewer2**
>
> We thank the reviewer for their extensive feedback. Principally, we want to emphasize that the goal of our work is neither to 1) outperform Vasic et al.’s proposed models, nor 2) propose models that train very quickly, although both of these are of course useful outcomes of our proposed techniques.
>
> The main contribution of our paper is the conceptual combination of structural features and global models. We leverage this idea to present two new models to embed code -- Sandwich models and GREAT -- which we compare against the current state-of-the-art in representation learning for code: GGNNs (Allamanis et al.). We show that our models perform significantly better not just in terms of convergence speed, but also in terms of final convergence accuracies (irrespective of the time the models take). Note that this improved performance is not just a product of switching from RNNs to (edge-biased) Transformers; the family of Sandwich models provides valuable insights into alternative forms of mixing message passing networks and traditional sequence-based models (RNN, Transformers) and are substantially more accurate with relatively few training samples. Indeed, some of our best results are achieved with RNN-style Sandwich models.
>
> Performance-wise, our proposed models do not just train faster, but are substantially more accurate regardless of training speed, both in terms of ultimate convergence, but especially also in terms of data efficiency. We recognize that our plots’ use of time on the x-axis may have given the impression that we were optimizing for efficiency given some time budget; so, we added an equivalent plot to Figure 1 that instead plots performance against the number of samples seen (conf. epochs), demonstrating that the various models’ curves are very similar regardless of time, and the convergent outcomes the same. This is also why we do not discuss our hardware (P100 GPUS, we will add this to the paper); our results are independent of time or implementation.
>
> Your comment regarding the risk of learning on synthetic data is well taken. Therefore, in addition to evaluating the models on synthetically generated bugs, we have also added an evaluation on variable-misuse errors mined from commits in a real-world GitHub dataset, in Section 5.5. The dataset was obtained by collecting one million commits to GitHub Python projects and extracting variable misuses using the same criteria we used to introduce them (i.e., a commit changing a single variable usage in a function, from one variable in scope to another), which we also used to introduce them synthetically. As can be observed from the table below, the GREAT and Sandwich models outperform GGNNs, Transformer, and BiRNN models by 7% or more.
>
> |Models| 		|Real-world GitHub Bug identification Accuracy|
> RNN                 		18.2%
> Transformer			22.8%
> GGNN				20.6%
> RNN Sandwich		31.6%
> Transformer Sandwich	27.1%
> GREAT 			27.7%
>
> Please find more detailed results on real-world GitHub bugs for different recall percentages in Table 2 in the updated draft.
>
> Regarding larger Transformer models: we considered transformers with up to 10 layers and 1024 attention dimension (substantially larger than those originally proposed by Vaswani et al.) and found that the relative difference in performance with our proposed model was consistent across architecture sizes (in terms of number of parameters). Furthermore, all our models in our various comparisons are matched across comparable ranges of parameters, regardless of the underlying architectures.
>
> It may indeed be beneficial to initialize a Transformer to incorporate edge-related bias, though we are not aware of any work that has proposed to do so. We suspect, however, that this will be less effective than incorporating the bias directly into the architecture, as we do. These two forms of inductive bias may also be complementary, which is worth investigating further.
>
> Your comment regarding ‘q’ and ‘k’ on page 3 is correct; we will improve this notation in the paper.
>
> We hope this helps clarify the novelty and significant improvements of our sandwich and GREAT models (in terms of final accuracy) over the previous state-of-the-art GGNN models. Please let us know if there are any other questions or comments.

---

> > ### Author Response · Authors · 2019-11-13
> > **Follow-up to AnonReviewer2**
> >
> > Please let us know if our response helped clarify your concerns regarding the synthetic training data's generalization, and helped elucidate our performance objectives. Also, please let us know about any additional clarifications or questions that may help improve your assessment of our work.

---

> > > ### Comment · AnonReviewer2 · 2019-11-13
> > > **Questions**
> > >
> > > Indeed, these new results are quite interesting.
> > >
> > > Could you confirm if the 170 commits do or do not have overlapping methods with the training data?
> > >
> > > Text clarifications:
> > > It is not clear from the text if the commits only change one token or there is only token change in the specific function (that just matches something changed elsewhere and was not a misuse).

---

> > > > ### Author Response · Authors · 2019-11-14
> > > > **Response to new questions**
> > > >
> > > > Thanks for your suggestion. We had earlier performed deduplication from Allamanis et al. (SPLASH’19)  in the train and test splits for our synthetic data results, but we constructed our real-world Github bug dataset independent of that. We have now additionally deduplicated our Github data w.r.t. our synthetic training data, which resulted in discovering a small amount of overlap -- 9 out of 170 bugs had similar functions in the training data. We have updated the results in the paper on this dataset with 161 bugs. As may be expected, all models’ performances dropped slightly, but the overall result remains the same; our models outperform prior models including GGNNs and sequence based models:
> > > >
> > > > |Models| 		|Real-world GitHub Bug localization & repair Precision|
> > > > RNN                 			13.3%
> > > > Transformer				15.8%
> > > > GGNN					17.7%
> > > > RNN Sandwich			28.6%
> > > > Transformer Sandwich	21.5%
> > > > GREAT 					23.7%
> > > >
> > > > To clarify the text on commits: we did not specifically limit our data to entire commits that only change one token; instead, we considered only functions changed in a commit if the commit changes a single token in that function, and that single-token change is an update to the name of a variable that is both declared and read in that function. Ensuring that the change only affects one of its usages, and that the variable is not declared out of the scope of that function avoids the risk of e.g. a global variable name change outside the function creating an apparent variable misuse. Additionally, since we are limiting to only a single token change in a function in which the same variable is used elsewhere at least once, this also removes cases in which there is e.g. a variable rename refactoring.
> > > >
> > > > The Adverse Effects of Code Duplication in Machine Learning Models of Code
> > > > M. Allamanis. SPLASH Onward! 2019

---

### Public Comment · ~Marc_Brockschmidt1 · 2019-10-10
**Questions**

Thank you for this paper. I was left with three questions:

(1) In the definition of your GREAT model, you define $b_{ij} = W_e^T \mathbf{e} + b_e$, where e is the type of the edge connecting $i$ and $j$. I don't understand the affine transformation here: Its result is fixed per edge-type, and hence could be "pre-computed" in the embedding of the edge type?

(2) There is concurrent work on improving transformers by making learned relations explicit submitted to ICLR (https://openreview.net/forum?id=B1xfElrKPr ). Could you comment on how their approach (which is an elementwise modulation of the output of the attention layer, based on the input) relates to your model?

(3) I am somewhat confused by the speed that you reported for your GNN implementation in section 5. Concretely, I've recently run _a lot_ of experiments on our publicly released VarMisuse dataset, with a number of different architectures. I can report that on a V100, a GGNN with 10 propagation steps and a hidden size of 128 converges in about 6 hours (processing about 120 graphs per second during training). While I understand that you use 16 rather than 10 propagation steps, I'm somewhat surprised that your models need 40x time to converge. Do you have intuition what causes this slowdown?

Thanks,
Marc

---

> ### Author Response · Authors · 2019-10-11
> **Response to Marc Brockschmidt's questions**
>
> Thank you for your interest in our work.
> (1) That is correct, these can be pre-computed per edge type; our implementation does in fact do this. We will improve the description in the paper.
>
> (2) Thank you for pointing out this work. The key difference in the model is that they introduce relations only after the usual MHA computations, on the resulting ‘values’, whereas our work directly alters the attention weights. As such, their approach does not include an explicit relational term, and does not explicitly take a sparse graph as input --- unlike our method. These two approaches seem quite orthogonal and could have complementary benefits; ours more closely follows the approach by Shaw et al. (2018).
>
> (3) First, we want to point out that both the GREAT and Sandwich models not only improve the training speed, but also significantly improve the final accuracy of the models. The GGNN models trained for significantly longer still perform significantly worse (>10% worse accuracy).
> Second, the dataset in Allamanis et al., ICLR 2018, is significantly smaller than ours. Our Python dataset contains more than 3 million samples. Prior work on this same Python dataset also found continued improvements over several days of training, albeit with RNNs (Vasic et al., ICLR 2019).
> Finally, our experience is similar to yours in the sense that we are able to get reasonable performance after 13 hours of training, but given our larger Python training set, we find that accuracy keeps increasing with much longer training times. Our reported time-to-convergence numbers are based on the time to best possible accuracy on a validation set.
> Our implementation gets similar throughput to what you describe. Our 8-layer GGNN’s with a 50K node batch-size trained at ~75 ‘full’ graphs/second, which increases to 150+ on graphs without AST nodes; these numbers become 50 & 100 for 16-layer GGNNs. However, the best-performing instances used a batch size of 12.5K, which train ca. 3x slower. We also note that we trained on P100 GPUs.
>
> We note that we could not evaluate our models on the dataset from Allamanis et al. (ICLR’18) because the dataset has incomplete sequence to graph node mapping information; the graphs in that work only included nodes that were reachable within 8 steps of a selected node, whereas we require the combined full graph (in alignment with the function body) for our hybrid models.

---

### Author Response · Authors · 2019-11-09
**New results on real-world bugs**

Several reviewers asked if our new models yield improved performance at detecting real world bugs. To explore this, we collected a set of 170 real-world variable misuse bugs from Github projects. The dataset contains both the original version of the code with a var misuse bug, and the fixed version.

We find that both of our new model classes, GREAT and sandwich, show a large practical improvement over any of the baseline models. See Table 2 in revised paper. For example, for classification accuracy (bug versus no-bug), we find:

RNN: 57.1%
Transformer: 62.9%
GGNN: 67.1%
RNN Sandwich: 67.7%
Trans. Sandwich: 76.5%
GREAT: 71.2%

Additionally, at localizing and repairing the bugs correctly, this is a much more difficult problem, but we similarly observe a large increase in performance:

RNN: 18.2%
Transformer: 22.8%
GGNN: 20.6%
RNN Sandwich: 31.6%
Transformer Sandwich: 27.1%
GREAT: 27.7%

---

### Decision · Program_Chairs · 2019-12-19

**Decision:**

Accept (Poster)

**Comment:**

The paper investigates hybrid NN architectures to represent programs, involving both local (RNN, Transformer) and global (Gated Graph NN) structures, with the goal of exploiting the program structure while permitting the fast flow of information through the whole program.

The proof of concept for the quality of the representation is the performance on the VarMisuse task (identifying where a variable was replaced by another one, and which variable was the correct one). Other criteria regard the computational cost of training and number of parameters.

Varied architectures, involving fast and local transmission with and without attention mechanisms, are investigated, comparing full graphs and compressed (leaves-only) graphs. The lessons learned concern the trade-off between the architecture of the model, the computational time and the learning curve. It is suggested that the Transformer learns from scratch to connect the tokens as appropriate; and that interleaving RNN and GNN allows for more effective processing, with less message passes and less parameters with improved accuracy.

A first issue raised by the reviewers concerns the computational time (ca 100 hours on P100 GPUs); the authors focus on the performance gain w.r.t. GGNN in terms of computational time (significant) and in terms of epochs. Another concern raised by the reviewers is the moderate originality of the proposed architecture. I strongly recommend that the authors make their architecture public; this is imo the best way to evidence the originality of the proposed solution.

The authors did a good job in answering the other concerns, in particular concerning the computational time and the choice of the samples. I thus recommend acceptance.